# Validation of serum galactomannan antigen assay for invasive pulmonary aspergillosis mortality outcome prediction

Trent Chang-Wei Wu,[1] Chen Chieh Lin,[1] Yung-Hsuan Chen,[2] Li-Ta Keng,[1] Lih-Yu Chang,[1] Jung-Yueh Chen,[3,4] Meng-Rui Lee,[1,2] Jann-Yuan Wang,[2] Chao-Chi Ho,[2] Jin-Yuan Shih[2]

**ABSTRACT** The galactomannan antigen assay is crucial in diagnosing invasive pulmonary aspergillosis (IPA). However, its clinical utility as a prognostic factor has not yet been fully validated. Patients with proven or probable invasive pulmonary aspergillosis (IPA) who had a serum galactomannan enzyme immunoassay optical density index (sGMI) (Platelia *Aspergillus* Antigen immunoassay, Bio-Rad, CA, USA) result between 2013 and 2020 at a tertiary referral center were identified. We conducted a systematic review and identified studies using static or kinetic sGMI markers for IPA prognosis. A multivariable logistic regression model was used to validate these sGMI markers separately for 30-day, 90-day, and in-hospital mortality. Our study included 268 IPA patients (14 proven IPA, 254 probable IPA). The 30-day, 90-day, and in-hospital mortality were 38.1%, 60.1%, and 61.6% respectively. A total of 12 different sGMI predictive markers were included for validation. In our multivariable logistic regression, baseline sGMI ≥2 at IPA diagnosis was associated with 30-day mortality (adjusted odds ratio [aOR] 2.06; 95% confidence interval [CI], 1.16–3.66; $P = 0.013$), 90-day mortality (aOR 2.33; 95% CI, 1.29–4.21; $P = 0.005$), and in-hospital mortality (aOR 2.99; 95% CI, 1.62–5.51; $P < 0.001$). A day 7 sGMI ≥1.5 was also associated with 30-day mortality (aOR 2.34; 95% CI, 1.09–5.02; $P = 0.029$), 90-day mortality (aOR 2.24; 95% CI, 1.10–4.58; $P = 0.027$), and in-hospital mortality (aOR 2.30; 95% CI, 1.12–4.71; $P = 0.023$). No kinetic sGMI marker reached statistical significance for predicting all three outcomes. A baseline sGMI ≥2 at diagnosis can be used as a prognostic marker for IPA mortality outcomes. A day 7 sGMI ≥1.5 during treatment serves as an additional prognostic marker.

**IMPORTANCE** Serum galactomannan enzyme immunoassay (sGMI) is a well-established, non-invasive diagnostic tool for invasive aspergillosis and has been incorporated into numerous clinical guidelines. Despite its diagnostic utility, the prognostic value of sGMI in IA patients was not well validated due to limitations in previous studies, including small sample sizes and methodological issues. This study systematically reviewed and validated existing sGMI prognostic markers in a large Taiwanese invasive pulmonary aspergillosis cohort. A sGMI ≥2 at baseline and a day 7 sGMI ≥1.5 were significant predictors for 30-day, 90-day, and in-hospital mortality. We propose that these markers can be used at different stages of IPA treatment to predict patient outcomes. Our research is the first to validate both static and kinetic sGMI markers in a large IPA cohort. The simplicity and practicality of using a baseline sGMI ≥2 at diagnosis for mortality prediction offer significant advantages and support medical decisions.

**KEYWORDS** invasive pulmonary aspergillosis, prognosis, galactomannan, survival analysis, biomarker

Invasive aspergillosis (IA) is a life-threatening disease affecting over 2 million patients worldwide, particularly immunocompromised individuals, such as those with

Address correspondence to Meng-Rui Lee, leemr@ntu.edu.tw.

The authors declare no conflict of interest.

hematologic malignancies, recipients of steroids or immunosuppressants, and critically ill patients in the intensive care unit (ICU) (1). The estimated annual mortality is as high as 85.2% with invasive pulmonary aspergillosis (IPA) being the most common and lethal form, accounting for over half of all IA cases (1). Given its high mortality rate, early and effective assessment of outcomes is crucial for guiding clinical management.

The galactomannan enzyme immunoassay (GM-EIA) has proven to be important in the diagnosis of invasive aspergillosis (IA) (2). GM-EIA can be applied to serum, bronchoalveolar lavage fluid, and even cerebrospinal fluid (3–5). Among these samples, the serum galactomannan enzyme immunoassay optical density index (sGMI) is the easiest, safest, and non-invasive approach, offering high specificity (6). As a result, it has been incorporated into various guidelines, being a fundamental part of mycological evidence for the diagnosis and monitoring of treatment response (7–12).

In addition to its diagnostic utility, the sGMI has also been proposed to have prognostic value. In previous studies, various static sGMI cutoffs or kinetic sGMI changes have been proposed as outcome predictors for IPA patients. However, these studies lack validation due to the small sample size of each study, univariate regression findings, and post-hoc studies from the early 2000s (13–24).

We initiated this study and conducted a systematic review to validate the literature-proposed sGMI prognostic markers in IPA patients in a large cohort from Taiwan. We expect to identify potential static or kinetic sGMI markers for predicting IPA mortality outcomes and aid physicians in better clinical management.

## MATERIALS AND METHODS

### Patient population and data collection

A retrospective cohort study was conducted at the National Taiwan University Hospital, a 3,000-bed tertiary medical center in Taipei, Taiwan. We included patients who were diagnosed with proven or probable IPA between January 1, 2013, and December 31, 2020. While we acknowledge that IPA is a diverse disease group with varying underlying diseases, we aimed to develop a model for generalizability in all IPA patients. We adopted the diagnostic criteria based on the latest 2020 EORTC/MSGERC definitions, along with the 2021 EORTC/(12MSGERC and 2024) FUNDICU definitions for critically ill patients (7, 8, 12). An sGMI cutoff of 0.5 for diagnosis was chosen for our cohort for several reasons: (i) previous studies in this field predominantly used the 0.5 cutoff in accordance with the manufacturer's and the 2008 EORTC/MSG definitions (25); (ii) the inclusion of a non-neutropenic and more immunocompetent patient population in our cohort would invariably lead to lower sGMI detected (26); and (iii) to unify the standards in the cohort of patients with hematologic malignancy and critical illness (7, 8, 12). The diagnosis of IPA was reviewed by two respiratory specialists, who evaluated clinical and radiological findings on the basis of CT, microbiological, and host criteria according to the abovementioned criteria.

Demographic data, underlying disease, serial sGMI measurements, immunosuppressant use, and radiographic findings were collected from electrical medical records. The sGMI was measured with the Platelia *Aspergillus* Antigen immunoassay (Bio-Rad, CA, USA). Risk factors for IA identified in previous studies were also included. The outcomes measured were 30-day mortality, 90-day mortality, and in-hospital mortality.

The study was approved by the Institutional Review Board of National Taiwan University Hospital (202008006RIND), with the requirement for written informed consent waived due to the retrospective nature of the study and the absence of risk to participants.

### Statistical analysis

Basic characteristics are presented as the means ± standard deviations or as numbers (percentages). Categorical variables were compared using the chi-square ($\chi^2$) test,

whereas continuous variables were analyzed using the independent *t*-test. There were missing sGMI data at subsequent follow-ups due to mortality before the designed sGMI follow-up, but no imputation or deletion was performed because of the nature of our study design.

A binary multivariable logistic regression model was developed to identify risk factors for 30-day, 90-day, and in-hospital mortality. We preselected clinically important variables, such as age, sex, BMI, neutropenic status, and antifungal treatment. Additionally, variables that retained statistical significance in univariate logistic regression, including hematologic malignancies, non-hematologic malignancies, hematopoietic stem cell transplantation (HSCT), solid organ transplant, immunosuppressant or steroid use, CT findings of consolidation, and ICU admission, were included in the final logistic regression model. A systematic review was conducted to select previously reported static and kinetic sGMI markers. We then included different sGMI markers in the final model along with the 12 variables mentioned above to validate their prognostic value. To highlight, only one sGMI marker is implemented each time into the model with the same combinations to avoid collinearity and overfitting. We then checked Spearman correlations and variance inflation factors among the variables in the model to avoid collinearity. The Kaplan–Meier analysis and receiver operating characteristic (ROC) curve area under the curve (AUC) were used to assess the performance and discrimination ability of the final model.

Data processing and statistical analyses were done using SPSS version 25.0 for Windows (IBM Corp., Armonk, NY, USA). The study adhered to the transparent reporting of studies developing multivariable prediction models for individual prognosis (TRIPOD) statement, as reported in Table S1. A two-sided *P* value of less than 0.05 was considered significant. The power was assumed to be 0.8, with half of the patients harboring the predictive marker and a 60% mortality rate. With an odds ratio of 3, the estimated sample size for the total cohort was 88 patients. All sGMI markers analyzed in our study surpassed the estimated number of required participants.

## RESULTS

### Cohort characteristics

Over the study period, we identified 1,378 patients with positive *Aspergillus* culture, GMI, or pathology results suggestive of pulmonary aspergillosis. A total of 300 patients met the diagnostic criteria for proven or probable IPA. Among them, 268 patients had at least one sGMI result, including 14 with proven IPA and 254 with probable IPA. The detailed recruitment process is illustrated in Supplementary Data Fig. S1.

The cohort characteristics are shown in Table 1. Among the 268 patients, there was a slight male predominance (*n* = 151, 56.3%) with a mean age of 56 years. The majority of the cohort had an underlying hematologic malignancy (*n* = 159, 59.3%) with acute myeloid leukemia (52/159, 32.7%), acute lymphoblastic leukemia (26/159, 16.4%), and myelodysplastic syndrome (23/159, 14.5%) being the most common diagnoses. Additionally, 45 patients (16.8%) had a non-hematologic malignancy, and 6 (2.2%) patients had a concomitant diagnosis of both hematologic and non-hematologic malignancy. A total of 85 patients (31.7%) had received transplantation, with 81 patients (30.2%) having undergone hematopoietic stem cell transplantation (HSCT) and six patients (2.2%) having received solid organ transplantation (three bilateral lung transplants, two renal transplants, and one liver transplant). A total of 46 patients (17.2%) had an autoimmune disease, and 124 patients (46.3%) were receiving immunosuppressant or steroid therapy at the time of IPA diagnosis. Neutropenic status, defined as an absolute neutrophil count <500/μL, was present in 123 patients (45.9%). There were 13 patients (4.9%) who had experienced severe influenza before the time of diagnosis, with no COVID-19 cases reported. Details of hematologic and non-hematologic malignancy, autoimmune disease, immunosuppressant, and solid organ transplant are listed in Table S2.

**TABLE 1** Demographic data of NTUH-IPA cohort[b,c,d]

| | All (n = 268) | IPA with underlying hematologic malignancy (n = 159) | IPA without underlying hematologic malignancy (n = 109) | P |
|---|---|---|---|---|
| Age | 56.58 ± 17.64 | 51.87 ± 16.71 | 63.45 ± 16.75 | **<0.001** |
| Male | 151 (56.3) | 86 (54.1) | 65 (59.6) | 0.369 |
| BMI | 22.26 ± 4.06 | 22.06 ± 4.00 | 22.54 ± 4.16 | 0.352 |
| Smoking | 66 (24.6) | 29 (18.2) | 37 (33.9) | **0.003** |
| Underlying diseases | | | | |
| Diabetes mellitus | 56 (20.9) | 26 (16.4) | 30 (27.5) | **0.027** |
| Hypertension | 85 (31.7) | 31 (19.5) | 54 (49.5) | **<0.001** |
| Human immunodeficiency virus | 2 (0.7) | 1 (0.6) | 1 (0.9) | 1.000 |
| Chronic lung disease | 29 (10.8) | 13 (8.2) | 16 (14.7) | 0.092 |
| Chronic obstructive pulmonary disease | 15 (5.6) | 6 (3.8) | 9 (8.3) | 0.117 |
| Asthma | 8 (3.0) | 3 (1.9) | 5 (4.6) | 0.277 |
| Old tuberculosis | 14 (5.2) | 6 (3.8) | 8 (7.3) | 0.197 |
| Cirrhosis of liver | 11 (4.1) | 5 (3.1) | 6 (5.5) | 0.363 |
| End-stage renal disease | 16 (6.0) | 5 (3.1) | 11 (10.1) | **0.018** |
| Autoimmune disease | 46 (17.2) | 5 (3.1) | 41 (37.6) | **<0.001** |
| Cancer (hematologic and non-hematologic malignancy) | 198 (73.9) | 159 (100.0) | 39 (35.8) | **<0.001** |
| Hematologic malignancy | 159 (59.3) | <u>159 (100.0)</u> | 0 (0.0) | |
| Non-hematologic malignancy | 45 (16.8) | 6 (3.8) | 39 (35.8) | **<0.001** |
| Organ transplant | 85 (31.7) | 76 (47.8) | 9 (8.3) | **<0.001** |
| HSCT | 81 (30.2) | 77 (48.4) | 4 (3.7) | **<0.001** |
| Allogeneic HSCT | 74 (27.6) | 70 (44.0) | 4 (3.7) | **<0.001** |
| Autologous HSCT | 7 (2.6) | 7 (4.4) | 0 (0.0) | **0.044** |
| Solid organ transplant | 6 (2.2) | 1 (0.6) | 5 (4.6) | **0.042** |
| Immunosuppressant or steroid use | 124 (46.3) | 69 (43.4) | 55 (50.5) | 0.255 |
| Immunosuppressant use | 92 (34.3) | 57 (35.8) | 35 (32.1) | 0.527 |
| Steroid use | 87 (32.5) | 41 (25.8) | 46 (42.2) | **0.005** |
| White blood cell count/µL | 6,836 ± 10,442 | 5,169 ± 10,784 | 9,254 ± 9,463 | **0.002** |
| Neutropenic status[a] | 123 (45.9) | 100 (62.9) | 23 (21.1) | **<0.001** |
| Post-influenza | 13 (4.9) | 2 (1.3) | 11 (10.1) | **0.001** |
| Proven IPA | 14 (5.2) | 7 (4.4) | 7 (6.4) | 0.465 |
| Probable IPA | 254 (94.8) | 152 (95.6) | 102 (93.6) | 0.465 |
| Aspergillus culture | 36 (13.4) | 9 (5.7) | 27 (24.8) | **<0.001** |
| *A. fumigatus* | 27 (10.1) | 6 (3.8) | 21 (19.3) | **<0.001** |
| *A. flavus* | 8 (3.0) | 2 (1.3) | 6 (5.5) | 0.066 |
| *A. fumigatus* and *A. flavus* | 1 (0.4) | 1 (0.6) | 0 (0.0) | 1.000 |
| Serum Galactomannan assay | | | | |
| Baseline sGMI | 2.43 ± 1.99 | 2.27 ± 1.94 | 2.67 ± 2.06 | 0.103 |
| Baseline sGMI ≥1 | 188 (70.1) | 109 (68.6) | 79 (72.5) | 0.491 |
| Baseline sGMI ≥2 | 110 (41.0) | 61 (38.4) | 49 (45.0) | 0.281 |
| Antifungal treatment | 237 (88.4) | 148 (93.1) | 93 (85.3) | 0.088 |
| Voriconazole | 190 (70.9) | 116 (73.0) | 74 (67.9) | 0.370 |
| Other medications | 47 (17.5) | 29 (18.2) | 18 (16.5) | 0.715 |
| No treatment | 31 (11.6) | 14 (8.8) | 17 (15.6) | 0.088 |
| Imaging finding | | | | |
| Consolidation | 183 (68.3) | 90 (56.6) | 93 (85.3) | **<0.001** |
| Nodules | 180 (67.2) | 114 (71.7) | 66 (60.6) | 0.056 |
| Halo sign | 96 (35.8) | 70 (44.0) | 26 (23.9) | **0.001** |
| Mass | 63 (23.5) | 33 (20.8) | 30 (27.5) | 0.199 |
| Cavitation | 38 (14.2) | 17 (10.7) | 21 (19.3) | 0.029 |
| Intensive care unit admission | 179 (66.8) | 93 (58.5) | 86 (78.9) | **<0.001** |

*(Continued on next page)*

**TABLE 1** Demographic data of NTUH-IPA cohort[b,c,d] (Continued)

| | All (n = 268) | IPA with underlying hematologic malignancy (n = 159) | IPA without underlying hematologic malignancy (n = 109) | P |
|---|---|---|---|---|
| 30-day mortality | 102 (38.1) | 57 (35.8) | 45 (41.3) | 0.368 |
| 90-day mortality | 161 (60.1) | 90 (56.6) | 71 (65.1) | 0.161 |
| In-hospital mortality | 165 (61.6) | 91 (57.2) | 74 (67.9) | 0.078 |

[a]Neutropenic status: defined as an absolute neutrophil count <500/μL.
[b]BMI, body mass index; IPA, Invasive pulmonary aspergillosis; HSCT, hematopoietic stem cell transplantation; sGMI, serum galactomannan enzyme immunoassay optical density index.
[c]P value <0.05: bold underlined.
[d]Data are presented in mean ± standard deviation or numbers (percentage) unless specified.

The most common radiologic findings were consolidations (68.3%), nodular lesions (67.2%), and lesions with a Halo sign (35.8%). The mean baseline sGMI was 2.43, with 188 (70.1%) patients having a value of two or greater. There were 36 (13.4%) patients who had positive fungal culture results from sputum, bronchial washing, or bronchoalveolar lavage samples. Among the positive isolates, *Aspergillus fumigatus* accounted for the majority (n = 27/36, 75.0%), followed by *Aspergillus flavus* (n = 8/36, 22.2%), and mixed *Aspergillus fumigatus* and *Aspergillus flavus* (n = 1/36, 2.7%). Antifungal medication was applied in 237 patients (88.4%), with first-line voriconazole prescribed in 190 patients (70.9%), and other treatments included echinocandins, amphotericin B, or posaconazole (n = 47, 17.5%). ICU care was required for 179 patients (66.8%) during hospitalization. The 30-day mortality, 90-day mortality, and in-hospital mortality rates were 38.1% (102/268), 60.1% (161/268), and 61.6% (165/268), respectively.

## sGMI markers validation

The detail of our systematic review is detailed in Fig. S2, Table S3 and Table S4. We validated 12 markers in our regression model for further comparison, including baseline sGMI, baseline sGMI ≥1.0, baseline sGMI ≥1.5, baseline sGMI ≥2, day 7 sGMI ≥1.5, the sGMI difference between day 7 and baseline (day 7 sGMI − baseline sGMI), the sGMI difference between day 14 and baseline (day 14 sGMI − baseline sGMI), the sGMI difference between day 14 and day 7 (day 14 sGMI − day 7 sGMI), the sGMI difference between the maximum and baseline (maximum sGMI − baseline sGMI), doubling in sGMI (100% increase from baseline), sGMI persistently ≥1.0 within 28 days, and an increasing trajectory of sGMI over 28 days.

## Univariate logistic regression analysis

The unadjusted ORs of all variables from our NTUH IPA cohort are listed in Supplementary Data Table 5. The 30-day, 90-day, and in-hospital mortality were more likely to occur in patients with a higher BMI, higher sGMI, CT findings of consolidation, hematologic malignancy without HSCT, and those who required ICU admission. A lower risk of mortality was observed in patients with hematologic malignancies who received HSCT and those who were on immunosuppressants.

## Multivariable logistic regression analysis

The final logistic regression model is shown in Table 2. The baseline sGMI was associated with higher mortality rate in all three outcomes 30-day mortality (adjusted odds ratio [aOR] 1.17; 95% confidence interval [CI], 1.02–1.35; P = 0.024), 90-day mortality (aOR 1.23; 95% CI, 1.05–1.45; P = 0.009), and in-hospital mortality (aOR 1.26; 95% CI, 1.07–1.49; P = 0.005).

Among the 12 selected sGMI markers, only two static sGMI markers were significantly associated with all three mortality outcomes, whereas the other static and kinetic markers were not associated. Baseline sGMI ≥2 was associated with a higher 30-day mortality (aOR 2.06; 95% CI, 1.16–3.66; P = 0.013), 90-day mortality (aOR 2.33; 95% CI, 1.29–4.21; P = 0.005), and in-hospital mortality (aOR 2.99; 95% CI, 1.62–5.51; P < 0.001).

**TABLE 2** Multivariable logistic regression model for mortality prediction: baseline serum galactomannan enzyme immunoassay optical density index ≥2 and day seven serum galactomannan enzyme immunoassay optical density index ≥1.5[a,b]

| | 30-day mortality | | 90-day mortality | | In-hospital mortality | |
|---|---|---|---|---|---|---|
| | aOR (95% CI) | *p* | aOR (95% CI) | *p* | aOR (95% CI) | *p* |
| Baseline sGMI ≥2 (*n* = 268) | | | | | | |
| Age (*n* = 268) | 1.00 (0.98–1.02) | 0.793 | 0.99 (0.97–1.01) | 0.170 | 0.99 (0.97–1.00) | 0.122 |
| Male (*n* = 151) | 0.89 (0.51–1.56) | 0.681 | 1.21 (0.69–2.14) | 0.503 | 0.76 (0.42–1.35) | 0.344 |
| BMI (*n* = 268) | 1.07 (1.00–1.15) | 0.050 | 1.11 (1.02–1.19) | **0.010** | 1.08 (1.00–1.17) | **0.046** |
| HSCT (*n* = 81) | 0.46 (0.20–1.07) | 0.071 | 0.39 (0.17–0.88) | **0.024** | 0.54 (0.23–1.26) | 0.153 |
| Solid organ transplant (*n* = 6) | 0.28 (0.03–2.83) | 0.284 | 0.07 (0.01–0.66) | 0.021 | 0.40 (0.07–-2.49) | 0.328 |
| Hematologic malignancy without HSCT (*n* = 82) | 1.01 (0.45–2.26) | 0.982 | 1.20 (0.51–2.80) | 0.675 | 0.81 (0.34–1.91) | 0.624 |
| Non-hematologic malignancy (*n* = 45) | 0.91 (0.41–1.99) | 0.808 | 1.15 (0.48–2.73) | 0.755 | 1.96 (0.78–4.92) | 0.153 |
| Immunosuppressant or steroid use (*n* = 124) | 0.54 (0.28–1.05) | 0.069 | 0.83 (0.43–1.60) | 0.571 | 0.70 (0.35–1.39) | 0.307 |
| Neutropenic status (*n* = 123) | 1.22 (0.66–2.27) | 0.523 | 1.14 (0.61–2.12) | 0.677 | 1.90 (1.00–3.60) | 0.050 |
| Consolidation pattern on CT (*n* = 183) | 1.77 (0.91–3.44) | 0.093 | 1.85 (0.98–3.50) | 0.057 | 2.14 (1.13–4.05) | **0.020** |
| Antifungal treatment (*n* = 237) | 0.38 (0.17–0.88) | **0.025** | 0.80 (0.32–1.97) | 0.621 | 0.82 (0.33–2.06) | 0.67 |
| ICU admission (*n* = 179) | 1.58 (0.84–2.98) | 0.153 | 2.12 (1.15–3.92) | **0.016** | 3.21 (1.71–6.02) | **<0.001** |
| Baseline sGMI ≥2 (*n* = 268) | 2.06 (1.16–3.66) | **0.013** | 2.33 (1.29–4.21) | **0.005** | 2.99 (1.62–5.51) | **<0.001** |
| Day 7 sGMI ≥1.5 (*n* = 169) | | | | | | |
| Age (*n* = 169) | 0.99 (0.97–1.02) | 0.543 | 0.99 (0.96–1.01) | 0.276 | 0.98 (0.96–1.01) | 0.208 |
| Sex (*n* = 93) | 0.64 (0.30–1.36) | 0.244 | 0.90 (0.43–1.86) | 0.776 | 0.57 (0.27–1.19) | 0.131 |
| BMI (*n* = 169) | 1.08 (0.98–1.18) | 0.113 | 1.17 (1.06–1.29) | **0.002** | 1.13 (1.03–1.25) | **0.013** |
| HSCT (*n* = 50) | 0.77 (0.23–2.53) | 0.666 | 0.69 (0.24–2.02) | 0.499 | 1.17 (0.39–3.50) | 0.777 |
| Solid organ transplant (*n* = 5) | 0.75 (0.06–9.69) | 0.827 | 0.19 (0.02–2.09) | 0.175 | 1.60 (0.22–11.78) | 0.647 |
| Hematologic malignancy without HSCT (*n* = 47) | 1.51 (0.50–4.54) | 0.463 | 1.29 (0.41–4.01) | 0.665 | 0.91 (0.29–2.85) | 0.866 |
| Non-hematologic malignancy (*n* = 32) | **1.02 (0.37–2.82)** | 0.977 | 1.10 (0.38–3.17) | 0.865 | 1.97 (0.66–5.87) | 0.225 |
| Immunosuppressant or steroid use (*n* = 80) | 0.35 (0.14–0.90) | **0.029** | 0.61 (0.25–1.50) | 0.279 | 0.52 (0.21–1.30) | 0.161 |
| Neutropenic status (*n* = 75) | 1.08 (0.47–2.48) | 0.859 | 0.92 (0.41–2.04) | 0.834 | 1.66 (0.74–3.74) | 0.218 |
| Consolidation pattern on CT (*n* = 120) | 2.90 (1.10–7.64) | **0.031** | 1.69 (0.74–3.89) | 0.217 | 2.35 (1.02–5.38) | **0.044** |
| Antifungal treatment (*n* = 154) | 0.37 (0.11–1.28) | 0.117 | 1.52 (0.43–5.34) | 0.512 | 1.28 (0.36–4.54) | 0.707 |
| ICU admission (*n* = 116) | 2.51 (0.99–6.34) | 0.052 | 2.10 (0.93–4.71) | 0.073 | 2.69 (1.19–6.11) | **0.018** |
| Day 7 sGMI ≥1.5 (*n* = 169) | 2.34 (1.09–5.02) | **0.029** | 2.24 (1.10–4.58) | **0.027** | 2.30 (1.12–4.71) | **0.023** |

[a]aOR, adjusted odds ratio; 95% CI, 95% confidence interval; HSCT, hematopoietic stem cell transplantation; sGMI, serum galactomannan enzyme immunoassay optical density index; ICU, Intensive care unit.
[b]P value < 0.05: bold underlined.

The other static marker, day 7 sGMI ≥1.5, was also significantly associated with 30-day mortality (aOR 2.34; 95% CI 1.09–5.02; *P* = 0.029), 90-day mortality (aOR 2.24; 95% CI, 1.10–4.58; *P* = 0.027), and in-hospital mortality (aOR 2.30; 95% CI, 1.12–4.71; *P* = 0.023).

The combination of baseline sGMI ≥2 and day 7 sGMI ≥1.5 was also associated with a higher 30-day mortality (aOR 2.27; 95% CI, 1.02–5.03; *P* = 0.044), in-hospital mortality (aOR 3.21; 95% CI, 1.34–7.67; *P* = 0.009), and borderline higher 90-day mortality (aOR 2.21; 95% CI, 0.99–4.91; *P* = 0.052). The Kaplan–Meier analysis revealed excellent discrimination with baseline sGMI ≥2, day 7 sGMI ≥1.5, and their combination, as shown in Fig. 1. The aOR comparison of the multivariable logistic regression model of the different sGMI markers is shown in Table 3.

The individual markers showed a fair performance in terms of ROC AUC as shown in Supplementary Data Fig. S3. The multivariable logistic regression model using baseline sGMI ≥2, a day 7 sGMI ≥1.5, and their combination had a good discrimination ability with ROC AUCs of 0.736, 0.769, and 0.767 for 30-day mortality; 0.745, 0.749, and 0.737 for 90-day mortality; and 0.752, 0.721, and 0.735 for in-hospital mortality, respectively, as shown in Supplementary Data Fig. S4.

# Kaplan Meier survival analysis for the parameters

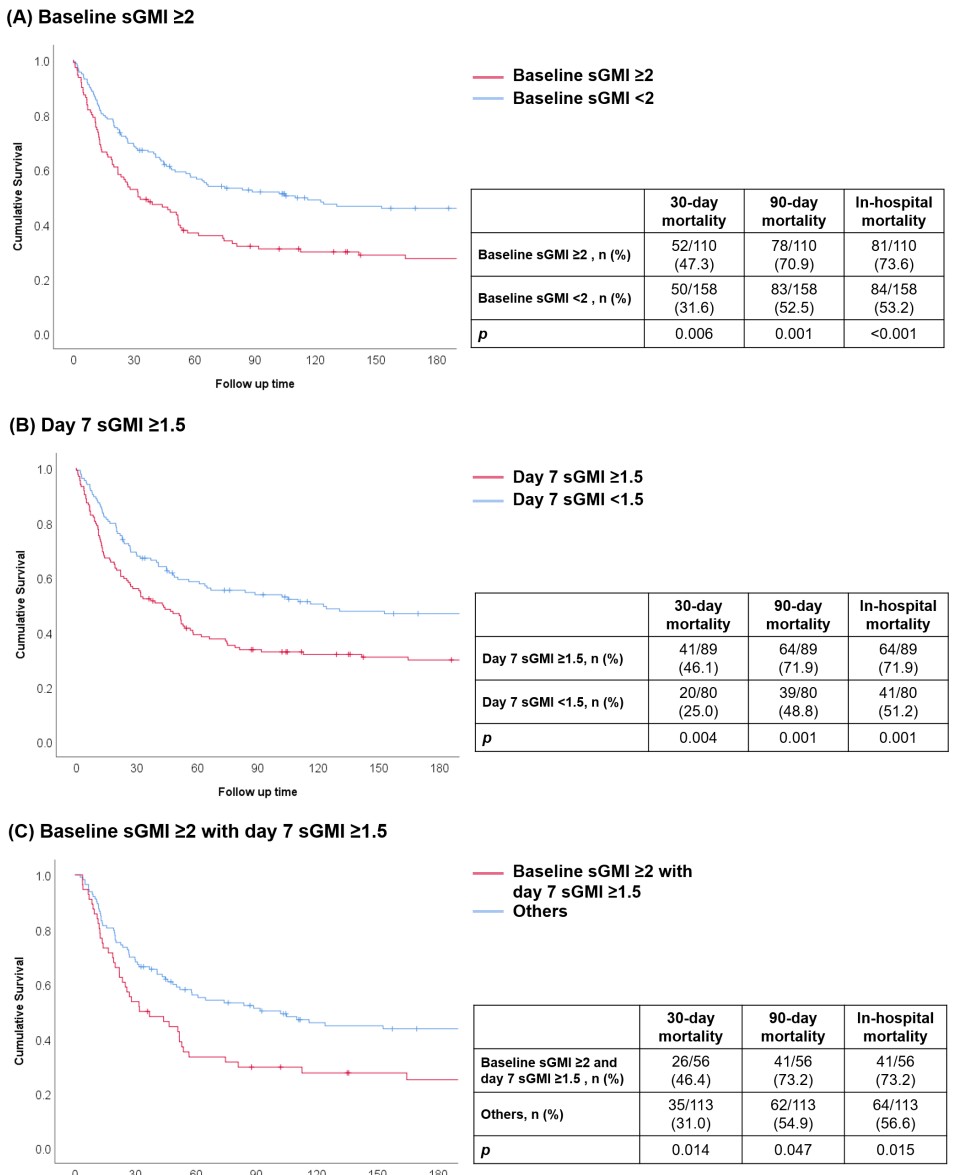

**FIG 1** Kaplan–Meier analysis of participants with validated sGMI markers. (A) Baseline sGMI ≥2, (B) Day 7 sGMI ≥1.5, (C) Baseline sGMI ≥2 with Day 7 sGMI ≥1.5. sGMI, serum galactomannan enzyme immunoassay optical density index.

## DISCUSSION

In our study, we have validated systematically reviewed sGMI prognostic markers in the NTUH-IPA cohort. As a result, a baseline sGMI ≥2, and a day 7 sGMI ≥1.5 were validated as prognostic markers for 30-day, 90-day, and in-hospital mortality. Taken together, a baseline sGMI ≥2 at the start of antifungal treatment and a day 7 sGMI (preferably with a cutoff of 1.5) could be used individually at different stages of IPA treatment to predict patient mortality outcomes.

Our study is the first to validate various static and kinetic sGMI markers in a large IPA cohort. Predicting mortality with a baseline sGMI marker is simpler and more practical than using kinetic markers, given the high mortality rate of IPA and the potential for

**TABLE 3** Validation of different serum galactomannan markers in the multivariable model[a,c,d]

|  | 30-day mortality | | 90-day mortality | | In-hospital mortality | |
| --- | --- | --- | --- | --- | --- | --- |
|  | aOR (95% CI) | p | aOR (95% CI) | p | aOR (95% CI) | p |
| Baseline sGMI [b] (n = 268) | 1.17 (1.02–1.35) | **0.024** | 1.23 (1.05–1.45) | **0.009** | 1.26 (1.07–1.49) | **0.005** |
| Baseline sGMI ≥1 | 1.72 (0.88–3.36) | 0.112 | 2.31 (1.21–4.43) | **0.012** | 2.71 (1.39–5.27) | **0.003** |
| Baseline sGMI ≥1.5 | 1.53 (0.87–2.71) | 0.142 | 2.00 (1.13–3.55) | **0.018** | 2.49 (1.38–4.49) | **0.002** |
| Baseline sGMI ≥2 [b] | 2.06 (1.16–3.66) | **0.013** | 2.33 (1.29–4.21) | **0.005** | 2.99 (1.62–5.51) | **<0.001** |
| Day 7 sGMI ≥1.5 [b] (n = 169) | 2.34 (1.09–5.02) | **0.029** | 2.24 (1.10–4.58) | **0.027** | 2.30 (1.12–4.71) | **0.023** |
| Day 7 sGMI − baseline sGMI (n = 169) | 1.12 (0.94–1.33) | 0.193 | 1.15 (0.97–1.37) | 0.111 | 1.08 (0.92–1.28) | 0.348 |
| Day 14 sGMI − baseline sGMI (n = 115) | 1.24 (0.98–1.59) | 0.078 | 1.16 (0.94–1.44) | 0.179 | 1.07 (0.87–1.31) | 0.550 |
| Day 14 sGMI − day 7 sGMI (n = 110) | 1.57 (1.03–2.39) | **0.036** | 0.99 (0.72–1.36) | 0.968 | 0.94 (0.68–1.29) | 0.706 |
| Maximum sGMI − baseline sGMI (n = 183) | 1.14 (0.90–1.45) | 0.267 | 1.32 (1.01–1.73) | **0.039** | 1.18 (0.93–1.51) | 0.174 |
| Doubling in sGMI (n = 183) | 1.00 (0.44–2.31) | 0.995 | 1.40 (0.62–3.17) | 0.426 | 1.46 (0.64–3.30) | 0.368 |
| sGMI persistently ≥1.0 (n = 183) | 5.71 (2.47–13.22) | **<0.001** | 2.47 (1.21–5.05) | **0.014** | 1.52 (0.76–3.07) | 0.238 |
| Increasing trajectory of sGMI within 28 days (n = 183) | 3.65 (1.68–7.94) | **0.001** | 2.06 (0.98–4.32) | 0.056 | 1.52 (0.74–3.13) | 0.254 |
| Baseline sGMI ≥2 with day 7 sGMI ≥1.5 (n = 169) | 2.27 (1.02–5.03) | **0.044** | 2.21 (0.99–4.91) | 0.052 | 2.47 (1.10–5.56) | **0.029** |

[a]Each sGMI marker was incorporated into the same multivariable logistic regression model separately. Only one sGMI marker was analyzed in the model each time.
[b]Statistically significant in 30-day mortality, 90-day mortality, and in-hospital mortality.
[c]aOR, adjusted odds ratio; 95% CI, 95% confidence interval; sGMI, serum galactomannan enzyme immunoassay optical density index.
[d]P value <0.05: bold underlined.

patients to die before day 7 or later sGMI follow-ups. Thus, the use of a baseline sGMI ≥2 at diagnosis offers clear advantages. Previous studies testing a baseline sGMI cutoff of two as a predictive marker have been limited, with only Fisher et al. reporting significant findings (17, 18, 27, 28). Our study, which includes a wider range of diagnoses beyond allogeneic HSCT patients, shows a similar mortality rate and enhances the generalizability of the prognostic value of baseline sGMI cutoff ≥2 (18, 28).

We attempted to validate all the markers identified in our systematic review within our model. A baseline sGMI with no cutoff was not practical despite its significant association with all three outcomes (13, 15, 29). In addition, we could not validate the cutoff of 0.5, the most investigated cutoff in previous studies, as all sGMI data in our cohort were above 0.5. However, recent guidelines have shifted the IPA diagnostic cutoff to 1, making the 0.5 cutoff less relevant for future care and studies outside of ICU (8). Other markers, such as sGMI differences between baseline and week 6, were also not applicable due to limited long-term sGMI follow-up in our retrospective design (14, 21). Therefore, we focused on the most practical and acceptable markers for our model comparison.

The study included 14 proven and 254 probable IPA cases. The trend of biopsy-proven IPA in the literature has lowered in recent years, likely due to the release of EORTC guidelines and popularization of the galactomannan test (7, 8). Moreover, 66.8% of the cohort patients experienced ICU admission. Critical illness often complicates with conditions such as respiratory failure and coagulopathy, which further limit the use of biopsy due to possible harm.

Our study has several notable strengths. First, we performed a comprehensive multivariable analysis and compared the prognostic efficacy of various markers using Kaplan–Meier analysis and ROC curve analysis, as illustrated in Fig. 1 and Supplementary Data Fig. S3 and S4. Second, rather than selecting subjective cutoff points, we conducted a systematic review to identify proposed cutoff points, which is a novel approach in this context. We evaluated multiple outcomes, including 30-day, 90-day, and in-hospital mortality. We did not assess treatment response according to the EORTC definition due to the lack of CT follow-up images in some patients (11) and the consideration of mortality being an objective endpoint for evaluation.

Despite previous studies suggesting that kinetic changes in the sGMI could serve as surrogate markers for treatment monitoring and mortality prediction, our study did not find them to be significant prognostic factors (22, 23, 29). Galactomannan is a polysaccharide in the cell wall of *Aspergillus* species, and as a pathogen-originated biomarker,

its level may be influenced by the degree of angioinvasion and the therapeutic drug levels of antifungal treatment, potentially not accurately reflecting the fungal burden throughout the treatment course (2). Furthermore, current evidence for sGMI markers primarily comes from studies with small sample sizes (<100), univariate regression findings, and post-hoc studies from the early 2000s (16, 22–24, 29). In our study, we validated these markers in a larger cohort with modern clinical care and employed a comprehensive multivariable logistic regression analysis.

Despite the lower sensitivity in non-neutropenic patients, GM-EIA remains one of the most objective markers for diagnosing IPA. It is a safer and more convenient approach compared to bronchoalveolar lavage or biopsy, especially for clinically suspected IPA patients, the majority of whom have hematologic malignancy or are critically ill. GM-EIA aids in the early suspicion and subsequent diagnostic process for IA. Newer techniques, such as plasma cell-free DNA, have shown promise in diagnosing and predicting invasive aspergillosis but still require further validation and broader adoption (17). The findings of our study indicate that the sGMI can serve as the simplest prognostic factor, helping physicians identify patients at high risk for mortality in addition to its diagnostic role.

However, there are still limitations to our study. Owing to the high mortality rate, it was challenging to obtain complete day 7, 14, and 28 sGMI results for prediction, leading to the exclusion of day 28 sGMI and kinetic changes due to a large amount of missing data. The traditional reliance on clinical and radiographic evidence for evaluating treatment response may have contributed to less frequent sGMI follow-ups and incomplete sequential data. Nevertheless, our study has the largest cohort for validating kinetic sGMI changes. Another limitation is the lack of other serum markers, such as beta-D-glucan, the galactomannan lateral flow assay, and *Aspergillus* PCR, as these are not diagnostic options available in our hospital at the time. A stringent review was performed by two respiratory specialists for IPA diagnosis confirmation. We are confident that the IPA diagnosis strictly adhered to the guidelines (7, 8, 12). Lastly, we did not assess treatment response in our study, which may have provided additional information to guide intervention. Notably, there was no COVID-19 infection or associated pulmonary aspergillosis in our cohort because there was no epidemic in Taiwan during the study period (30). In conclusion, a baseline sGMI of 2 or greater at IPA diagnosis, and day 7 sGMI of 1.5 or greater are independently associated with 30-day, 90-day, and in-hospital mortality. Kinetic sGMI markers are not reliable prognostic markers. Our findings offer a straightforward and feasible method for predicting mortality in IPA patients.

## ACKNOWLEDGMENTS

This research received no specific grant from any funding agency in the public, commercial, or not-for-profit sectors. This study was published as a pre-print version in *Research Square* (31).

T.C.-W.W. contributed to the analysis and interpretation of data and drafting the article. C.C.L. contributed to the acquisition, analysis, and interpretation of data. Y.-H.C. contributed to the acquisition, analysis, and interpretation of data. L.-T.K. contributed to the acquisition, analysis, and interpretation of data. L.-Y.C. contributed to the acquisition, analysis, and interpretation of data. J.-Y.C. contributed to the analysis and interpretation of data. M.-R.L. contributed to the conception and design of the study, acquisition of data, analysis and interpretation of data, revising the article critically for important intellectual content, and the final approval of the version to be submitted. J.-Y.W. contributed to the conception and design of the study and revising the article. C.-C.H. contributed to the conception and design of the study and revising the article. J.-Y.S. contributed to the conception and design of the study and revising the article. All the authors have read and approved the final manuscript.

## AUTHOR AFFILIATIONS

[1]Division of Pulmonary and Critical Care Medicine, Department of Internal Medicine, National Taiwan University Hospital Hsin-Chu branch, Hsin-Chu City, Taiwan

²Division of Pulmonary and Critical Care Medicine, Department of Internal Medicine, National Taiwan University Hospital, Taipei, Taiwan

³School of Medicine, College of Medicine, I-Shou University, Kaohsiung City, Taiwan

⁴Department of Internal Medicine, E-DA Hospital, I-Shou University, Kaohsiung, Taiwan

## AUTHOR ORCIDs

Trent Chang-Wei Wu ⓘ http://orcid.org/0000-0003-4838-9396

Meng-Rui Lee ⓘ http://orcid.org/0000-0002-7220-4833

Jann-Yuan Wang ⓘ http://orcid.org/0000-0003-3406-366X

## AUTHOR CONTRIBUTIONS

Trent Chang-Wei Wu, Data curation, Formal analysis, Writing – original draft, Writing – review and editing | Chen Chieh Lin, Data curation, Formal analysis | Yung-Hsuan Chen, Data curation, Formal analysis | Li-Ta Keng, Data curation, Formal analysis | Lih-Yu Chang, Data curation, Formal analysis | Jung-Yueh Chen, Formal analysis | Meng-Rui Lee, Conceptualization, Data curation, Formal analysis, Supervision, Writing – review and editing | Jann-Yuan Wang, Conceptualization, Supervision, Writing – review and editing | Chao-Chi Ho, Conceptualization, Supervision, Writing – review and editing | Jin-Yuan Shih, Conceptualization, Supervision, Writing – review and editing

## DATA AVAILABILITY

The data for this study can be obtained from the corresponding authors upon reasonable request.

## ADDITIONAL FILES

The following material is available online.

### Supplemental Material

**Supplemental materials (Spectrum00651-25-s0001.docx).** Tables S1 to S5, Figures S1 to S4, and Supplemental information S1 and S2.

### Open Peer Review

**PEER REVIEW HISTORY (review-history.pdf).** An accounting of the reviewer comments and feedback.

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
