## [Reviewer comments · Microbiology Spectrum]

Microbiology Spectrum

Validation of Serum Galactomannan Antigen Assay for Invasive Pulmonary Aspergillosis Outcome Prediction

Trent Chang-Wei Wu, Chen-Chieh Lin, Yung-Hsuan Chen, Li-Ta Keng, Lih-Yu Chang, Jung-Yueh Chen, Meng-Rui Lee, Jann-Yuan Wang, Chao-Chi Ho, and Jin-Yuan Shih

Corresponding Author(s): Meng-Rui Lee, National Taiwan University Hospital

Review Timeline:

Submission Date:	April 2, 2025
Editorial Decision:	August 19, 2025
Revision Received:	September 27, 2025
Accepted:	September 30, 2025

Editor: Po-Yu Liu

Reviewer(s): The reviewers have opted to remain anonymous.

Transaction Report:

DOI: <https://doi.org/10.1128/spectrum.00651-25>

Re: Spectrum00651-25 (Validation of Serum Galactomannan Antigen Assay for Invasive Pulmonary Aspergillosis Outcome Prediction)

Dear Dr. Meng-Rui Lee:

Thank you for the privilege of reviewing your work. Below you will find my comments, instructions from the Spectrum editorial office, and the reviewer comments.

In addition to the specific points from the reviewers listed below, there is a critical editorial requirement for revision. As discussed internally, to align your manuscript with Spectrum's focus on original research articles, please restructure the paper to unequivocally present it as an original study. The systematic review component should be framed strictly as a methodological step for variable selection, and the detailed PRISMA methodology and related figures/tables should be moved to the supplementary materials. The main text should focus on the original data, analysis, and validation findings from your patient cohort.

Revision Guidelines

Sincerely,
Po-Yu Liu
Editor
Microbiology Spectrum

Reviewer #1 (Comments for the Author):

I am not sure about the specific length of hospital stay for these patients, but it seems that the in-hospital mortality rate is higher than the 90 day mortality rate, which means that the average length of hospital stay is greater than 90 days? This does not conform to common sense.

Reviewer #2 (Comments for the Author):

1. The study focused on the relationship of galactomannan sGMI and short-term mortality. There was only 36 (13.4%) patients who had positive fungal culture results. what is the diagnostic criteria of the other subjects? In the methods section, it would be helpful to specify the diagnostic criteria of IPA, including proven or probable IPA. Although an sGMI cutoff of 0.5 for diagnosis was chosen for the cohort, are there any factors affected the results of galactomannan sGMI, which may cause false positive because of some medicine or albumin treatment?

2. In Table 1, the total number of Cancer (hematologic and non-hematologic malignancy) is 198, however, it showed 159 Hematologic malignancy and 45 Non-hematologic malignancy. The same question about the numbers of "Immunosuppressant or steroid use", with 124 (92 + 87?)

The data in Table 2a should also be checked again, eg. Immunosuppressant or steroid use (n=124)?

1. The study focused on the relationship of galactomannan sGMI and short-term mortality. There was only 36 (13.4%) patients who had positive fungal culture results. what is the diagnostic criteria of the other subjects? In the methods section, it would be helpful to specify the diagnostic criteria of IPA, including proven or probable IPA. Although an sGMI cutoff of 0.5 for diagnosis was chosen for the cohort, are there any factors affected the results of galactomannan sGMI, which may cause false positive because of some medicine or albumin treatment?

2. In Table 1, the total number of Cancer (hematologic and non-hematologic malignancy) is 198, however, it showed 159 Hematologic malignancy and 45 Non-hematologic malignancy. The same question about the numbers of “Immunosuppressant or steroid use”, with 124 (92 + 87?)

The data in Table 2a should also be checked again, eg. Immunosuppressant or steroid use (n=124)?

Response to Reviewer Comments

Critical editorial requirement for revision.

As discussed internally, to align your manuscript with Spectrum's focus on original research articles, please restructure the paper to unequivocally present it as an original study. The systematic review component should be framed strictly as a methodological step for variable selection, and the detailed PRISMA methodology and related figures/tables should be moved to the supplementary materials.

Response:

Adjustment were made according to request. We have restructured the paper as a original research article and framed the systematic review as a methodological step only.

Only minimal information of the systematic review was remained in the main article to justify our selection of sGMI markers. All details were moved to Supplementary Data Figure 2, Supplementary Data Table 3, Supplementary Data Table 4.

Page 10, Row 143-144

A systematic review was conducted to select previous reported static and kinetic sGMI markers.

Page 14, Row 203-213

sGMI Markers Validation

The detail of our systematic review is detailed in Supplementary Data Figure 2, Supplementary Data Table 3, Supplementary Data Table 4. We validated 12 markers in our regression model for further comparison, including baseline sGMI, baseline sGMI ≥ 1.0 , baseline sGMI ≥ 1.5 , baseline sGMI ≥ 2 , day 7 sGMI ≥ 1.5 , the sGMI difference between day 7 and baseline (day 7 sGMI – baseline sGMI), the sGMI difference between day 14 and baseline (day 14 sGMI – baseline sGMI), the sGMI difference between day 14 and day 7 (day 14 sGMI – day 7 sGMI), the sGMI difference between the maximum and baseline (Maximum sGMI – baseline sGMI), Doubling in sGMI (100% increase from baseline), sGMI persistently ≥ 1.0 within 28 days, and an increasing trajectory of sGMI over 28 days.

Reviewer #1

Reviewer #1 (Comments for the Author):

I am not sure about the specific length of hospital stay for these patients, but it seems that the in-hospital mortality rate is higher than the 90 day mortality rate, which means that the average length of hospital stay is greater than 90 days? This does not conform to common sense.

Response:

Thank you for your critical opinion. Limited data were reported on IPA related in-hospital length so we are unable to compare. The 90-day mortality was higher in our cohort compared with previous literature, which was around 30-40%. This may be related to our complicated underlying disease population (59.3% of the cohort had hematologic malignancy, 31.7% received hematopoietic stem cell or solid organ transplantation, 16.8% had non-hematologic malignancy). Moreover, 66.8% of the cohort patients experienced ICU admission. This may have resulted in a longer duration of hospitalization. Additionally, Taiwan has a population-based single-payer National Health Insurance system which is known for its convenience and low cost for patients and family. Lengthy hospitalization more than 90 days, especially among complicated patients, are not uncommon in Taiwan hospitals.

Reviewer #2 (Comments for the Author):

1. The study focused on the relationship of galactomannan sGMI and short-term mortality. There was only 36 (13.4%) patients who had positive fungal culture results. what is the diagnostic criteria of the other subjects? In the methods section, it would be helpful to specify the diagnostic criteria of IPA, including proven or probable IPA. Although an sGMI cutoff of 0.5 for diagnosis was chosen for the cohort, are there any factors affected the results of galactomannan sGMI, which may cause false positive because of some medicine or albumin treatment?

Response:

Thank you for your excellent opinion.

1. There was only 36 (13.4%) patients who had positive fungal culture results?

There were 14 proven and 254 probable IPA cases included in the study. The trend of biopsy proven IPA in literatures has lowered in recent years, likely related to the release of EORTC guidelines and popularization of galactomannan test. For instance, in the study by Marr et al (*Ann Intern Med.* 2015;162:81-9. doi: 10.7326/M13-2508), only 1.8% were proven IA and 78.7% of the study population had no histopathology or culture evidence.

Moreover, 66.8% of the cohort patients experienced ICU admission. Critical illness often complicates with conditions such as respiratory failure, coagulopathy and further limits the use of biopsy or even bronchoscopy due to possible harm.

2. What is the diagnostic criteria of the other subjects?

94.8% of our IPA cohort were probable IPA. We adopted the 2020 EORTC/MSGERC definitions for non-critical ill patients, along with the 2021 EORTC/MSGERC and 2024 FUNDICU definitions for critical ill patients. Due to the complexity, we highlighted the key spirit of criteria (Page 8, line 107-118): guideline we adhered, sGMI cutoff, clinical, radiological findings on the basis of CT, microbiological, and host criteria.

3. False positive because of some medicine or albumin treatment?

This is indeed difficult to identify. Albumin, beta-lactams, and total parenteral nutrition are some of the previous identified culprit drug that causes false positivity in sGMI. While reviewing the diagnosis criteria, two of our pulmonologist took into account of these drugs. Moreover, all cases were diagnosed with a computed tomography with compatible IPA lesions, 183 of the patients received 2 or more galactomannan test, which further supports the confidence of our diagnosis. However, Shin et al. reported glucose containing solution as a culprit drug (*Sci Rep.* 2024 Jan 31;14(1):2552. doi: 10.1038/s41598-024-53116-x). 66.8% of our cohort patients experienced ICU admission and was used in many critical ill patients and

difficult to be retrospectively identified.

2. In Table 1, the total number of Cancer (hematologic and non-hematologic malignancy) is 198, however, it showed 159 Hematologic malignancy and 45 Non-hematologic malignancy. The same question about the numbers of "Immunosuppressant or steroid use", with 124 (92 + 87?)

The data in Table 2a should also be checked again, eg. Immunosuppressant or steroid use (n=124)?

1. Number difference in hematologic malignancy/ Immunosuppressant or steroid use?

Thank you for your important question. There was an overlap in both categories which explains the difference. 6 patients had both hematologic and non-hematologic malignancy (Page 11, Row 176-178). 45 patients were under both immunosuppressant and steroid.

Re: Spectrum00651-25R1 (Validation of Serum Galactomannan Antigen Assay for Invasive Pulmonary Aspergillosis Outcome Prediction)

Dear Dr. Meng-Rui Lee:

Your manuscript has been accepted, and I am forwarding it to the ASM production staff for publication. Your paper will first be checked to make sure all elements meet the technical requirements. ASM staff will contact you if anything needs to be revised before copyediting and production can begin. Otherwise, you will be notified when your proofs are ready to be viewed.

Sincerely,
Po-Yu Liu
Editor
Microbiology Spectrum